# Large Language Models as Planning Domain Generators

**Primary Keywords:** *None*

## Abstract

The creation of planning models, and in particular domain models, is among the last bastions of tasks that require extensive manual labor in AI planning; it is desirable to simplify this process for the sake of making planning more accessible. To this end, we investigate whether large language models (LLMs) can be used to generate planning domain models from textual descriptions. We propose a novel task for this as well as a means of automated evaluation for generated domains by comparing the sets of plans for domain instances. Finally, we perform an empirical analysis of 7 large language models, including coding and chat models across 9 different planning domains. Our results show that LLMs, particularly larger ones, exhibit some level of proficiency in generating correct planning domains from natural language descriptions.

## 1 Introduction

Large language models (LLMs) have demonstrated robust emergent abilities for open-ended tasks like story generation, poetry, and dialogue (Zhao et al. 2023b; Hayawi, Shahriar, and Mathew 2023). Their potential is no longer limited to natural language. Rather, they have shown the ability to generate highly structured output that resembles code from natural language descriptions of programs (Li, Allal, and Zi 2023; Touvron, Lavril, and Izacard 2023). It is natural to wonder how these abilities generalize to knowledge engineering tasks such as those used for problem representation in symbolic methods. Despite the efficacy of symbolic methods such as boolean satisfiability (SAT) solvers (Biere et al. 2021), automated planners (Helmert 2006), and automated theorem provers (Harrison, Urban, and Wiedijk 2014) in their respective domains, the issue of representing a problem accurately and efficiently still hinders the wider adoption and accessibility of these powerful methods. If LLMs can bridge the gap between natural language description of the problem and symbolic representation, it would enable large-scale adoption of symbolic methods and reduce the dependency on technical experts. Motivated by this, we investigate LLMs for generating problem representations for automated planning (Ghallab, Nau, and Traverso 2004). We explore whether the commonsense knowledge, natural language capabilities, and emergent structured code generation ability of LLMs help constructing declarative planning domains. Specifically, we leverage LLMs to automatically translate natural language description of a domain to Planning Domain Description Language (PDDL) (Fox and Long 2003).

The problem of domain generation from natural language has been studied earlier (Lindsay et al. 2017; Hayton et al. 2020) and recently Guan et al. (2023) also attempted this problem using LLMs. Despite these studies, the task of evaluating the usefulness of the generated domain description is extremely difficult. Previous works leveraged human experts for evaluation. We argue that for rigorous, automated evaluation we need a ground truth; a vetted domain specification. Hence, in this work we focus on the task of creating *high quality reconstructions* of PDDL domain from natural language; where the generated domain is ideally equivalent to the ground truth. Restricting the generation of PDDL domain to an approximated equivalence class would make the generated domains more amenable to existing planners and further the goal of using the generated descriptions for producing executable plans. To further clarify, while Guan et al. (2023) uses LLMs to learn a PDDL models from a textual descriptions, this is not our main purpose in this work. We aim at in this work to understand how such methods can be evaluated, and due to this, need to depend on additional assumption that a reference domain is available. While this is a stronger assumption than what is made in earlier work, this allows for fully automated evaluation.

The core contributions of this work are fourfold. First, we define a task of PDDL domain reconstruction from natural language; based off a ground truth. Second, we define two metrics for evaluating domain quality that do not depend on any form of manual human evaluation. Third, we examine classes of natural language descriptions of PDDL actions to investigate if the inclusion and exclusion of particular information impacts the ability to generate domains or the quality of generated domains. Finally, we evaluate 7 different LLMs, including coding and chat models, and provide a detailed analysis of the results from each on 9 domains.

## 2 Background

### Planning

There are many ways to represent planning problems; formalisms over the years have been driven by concerns of efficiency and accessibility. Some of these plan representations

include the STRIPS (Ghallab et al. 1998), ADL (Pednault 1994), and SAS+ (Bäckström and Nebel 1995) formalisms.

In this work, we use the Planning Domain Definition Language (PDDL) for the declarative plan representation, but when necessary to discuss the underlying formalisms we refer to parts of planning problems and domains using the following lifted STRIPS formalism, largely inline with Corrêa and Seipp (2022). A lifted STRIPS planning problem is defined as a 5-tuple $\Pi = \langle \mathcal{F}, \mathcal{C}, \mathcal{A}, s_0, S_* \rangle$. $\mathcal{F}$ is a finite set of predicates that describe the world. $\mathcal{C}$ is a finite set of constants representing objects in the world, optionally including type information. We define $\mathcal{F}_g$ as the set of all *grounded predicates*, that is, predicates in which all variables are replaced by legal constants from $\mathcal{C}$. A state $s \subseteq \mathcal{F}_g$ is a set of grounded predicates that describe the state of the world, such that $f \in s$ if and only if $f$ is a true fact about the world. The set of all possible states is the power set of $\mathcal{F}_g$, denoted by $S$. $\mathcal{A}$ is a set of action schema where each $a \in \mathcal{A}$ is a 3-tuple $\langle pre(a), add(a), del(a) \rangle$ where $pre(a) \subseteq \mathcal{F}$ is the set of predicates that must hold to apply the action, $add(a) \subseteq \mathcal{F}$ is the set of predicates that become true after the action is applied, and $del(a) \subseteq \mathcal{F}$ is the set of predicates that become false. An action schema $a \in A$ can be grounded by substituting all variables in $a$ with allowed constants from $\mathcal{C}$. The grounded action $a_g = \langle pre_g(a_g), add_g(a_g), del_g(a_g) \rangle$ is defined as a 3-tuple of its grounded $pre$, $add$, and $del$ predicates, and we define $\mathcal{A}_g$ as the set of all grounded actions. Finally, $s_0 \subseteq \mathcal{F}_g$ is the initial state of the world for the planning task and $S_* \subseteq S$ is the set of possible goal states.

For a grounded action $a_g \in \mathcal{A}_g$ and a state $s \in S$, we say that $a_g$ is *applicable* in $s$ if $pre_g(a_g) \subseteq s$. Applying an applicable action $a_g$ in the state $s$ results in a state $s[a_g] := (s/del_g(a_g)) \cup add_g(a_g)$. A plan for a problem $\Pi$ is therefore a sequence of grounded actions $\pi = (a_1, \cdots, a_n)$ which when applied transforms the initial state $s_0$ into a goal state in $S_*$. The action sequence defines a state sequence $\mathbf{S} = (s_0, \cdots, s_n)$ such that $s_i = s_{i-1}[a_i]$ for $1 \leq i \leq n$ and $s_n \in S_*$. The set of all plans for $\Pi$ is denoted by $\mathcal{P}_\Pi$.

A *planning domain* for a lifted STRIPS planning problem $\Pi$ is the problem's predicate and action schema sets $\mathbf{D} = \langle \mathcal{F}, \mathcal{A} \rangle$, while we say $\Pi$ is a problem for $\mathbf{D}$ and write $\Pi_{\mathbf{D}}$ if $\Pi$ uses $\mathbf{D}$ as its underlying domain, regardless of the specific objects, initial state, and goal states $(C, s_0, S_*)$ for the problem.

## Large Language Models (LLMs)

*Language Models* are probabilistic predictors for language tokens that when given a sequence of tokens $T = (t_0, t_1, \cdots, t_n)$ in a corpus $C$ will output a set of predictions and associated probabilities $P \subseteq C \times \mathbb{R}$ for $t_{n+1}$ based on the data the model has been trained on. Different *decoding strategies* can be used to select a token in $P$ based on the probabilities, one such strategy is the greedy strategy which sets $t_{n+1}$ equal to the highest probability token in $P$. The new $t_{n+1}$ can be appended to $T$ and the process can be repeated for the next token. The maximum allowed size of $T$ is known as the *context window*, which limits the amount of tokens able to use for prediction.

*Large language models* are a class of language model characterized by their large size and emergent abilities on tasks that smaller language models are unable to perform on. LLMs are almost always implemented on-top of a Transformer architecture (Vaswani et al. 2017). There are many different types of large language models trained on various types of data, and models may be tuned to perform different types of tasks such as code generation (Li, Allal, and Zi 2023) or acting as chat agents (Touvron, Lavril, and Izacard 2023); a survey can be found at (Zhao et al. 2023a).

*In-Context Learning* for LLM inference is a technique classified as an emergent ability of LLMs to perform at a higher level of performance on tasks using examples of desired inputs and outputs (Dong et al. 2022). For example, rather than the prompt: "`Solve the following addition problem: 1 + 2`", an in-context learning prompt would read: "`Solve the following addition problems: In: 2 + 3, Out: 5; In: 4 + 2, Out: 6; ..., In: 1 + 2,`", where the prompt is composed of 3 parts (1) An *instruction* (2) a set of *context examples* and a (3) a *query* which is expected to be answered inline with the context examples. In-context learning is used in our work and much of the related work such as (Liu et al. 2023) and (Guan et al. 2023).

# 3 Approach

The goal of this work is the evaluation of LLM's abilities to generate PDDL domains. In particular, we are interested in generating and evaluating these domains on an action-by-action basis where each prompt to the LLM is a request to generate one action in a domain using context examples from other domains. This action-by-action prompting was inspired by Guan et al. (2023) and is primarily a concern due to the size of the LLM's context window.

We now turn to characterizing the concrete task we are trying to solve, an overview can be seen in Figure 1. In order to evaluate generated domains automatically, a ground truth domain is needed to compare the generated domains against. For this we use existing PDDL domains as a starting point in our approach. Given a starting domain $\mathbf{D} = \langle \mathcal{F}, \mathcal{A} \rangle$, we begin by converting all action schema in $\mathcal{A}$ to natural language descriptions of action schema, $N(\mathcal{A})$. We assume that a list of the predicates in the domain $\mathcal{F}$ and natural language descriptions of these predicates $N(\mathcal{F})$ are given to us as context for the domain. This assumption, while slightly limiting accessibility, is the cornerstone which allows this task a much more robust set of automatic evaluation options than when the context for the domain is just a natural language description (as in Guan et al. (2023)). The natural language action $N(a) \in N(\mathcal{A})$, along with a specification of domain predicates $\langle \mathcal{F}, N(\mathcal{F}) \rangle$, is used as the query for the in-context learning prompt. For the prompt's context examples, other actions are randomly sampled from action schema outside of the domain $\mathbf{D}$ of the current action. A model then takes these prompts and transforms them into a sequence of tokens $T(a)$ representing $a$ as a PDDL action. An attempt is made to parse $T(a)$ as a PDDL action $a'$. This is the first location at which automated evaluation is possible, as there are numerous reasons why $T(a)$ may fail to be a valid PDDL action, many of which can be extracted by just attempting

Listing 1: A context example from a prompt $N(a)$ for the fly-airplane action from the logistics domain, including the "Allowed Predicates" which function as the domain specification $\langle \mathcal{F}, N(\mathcal{F}) \rangle$.

```
Allowed Predicates:
(in-city ?loc - place ?city - city) : a
    place loc is in a city.
(at ?obj - physobj ?loc - place) : a
    physical object obj is at a place loc.
(in ?pkg - package ?veh - vehicle) : a
    package pkg is in a vehicle veh.

Input:
The action, "FLY-AIRPLANE" will fly an
    airplane from one airport to another.
    After the action, the airplane will be
    in the new location.

PDDL Action:
(:action FLY-AIRPLANE
    :parameters (?airplane - airplane ?loc-
        from - airport ?loc-to - airport)
    :precondition (at ?airplane ?loc-from)
    :effect (and (not (at ?airplane ?loc-
        from)) (at ?airplane ?loc-to))
)
```

to parse $T(a)$. For all $T(a)$ that were successfully parsed into a reconstructed PDDL action $a'$, we add them to the set of successfully reconstructed actions $\mathcal{A}'$. Next, for each $a' \in \mathcal{A}'$ we create a reconstructed domain $\mathbf{D}'$ from $\mathbf{D}$ by replacing $\mathcal{A}$ with $(A/a) \cup a'$ where $a$ is the original action that generated $a'$. Note that for our formulation $\mathcal{A}'$ is not the set of actions for a $\mathbf{D}'$, rather we look at $|\mathcal{A}'|$ new domains $\mathbf{D}'$s for each action, inline with our action by action-based evaluation strategy. This is also due to practicality reasons, in order to use $\mathcal{A}'$ for $\mathbf{D}'$, all actions in the domain would need to get through the parsing phase in which $T(a)$ is converted to $a'$, this is simply not a reasonable assumption to make. Our task then, is to evaluate the quality of each $\mathbf{D}'$ with respect to $\mathbf{D}$.

### Description Classes

We investigate several strategies for converting PDDL action schema $a \in \mathcal{A}$ to their natural language descriptions, $N(a) \in N(\mathcal{A})$. Each strategy produces a distinct class of natural language representations of the action model.

1. **Base** $N_b(\mathcal{A})$: Base descriptions include only information including the action name, parameters, and the parameter types of the action, as well as a one-line description of what the action does without explicitly mentioning any predicates. For example: *"The action 'unstack' will have a hand unstack a block x from a block y."*

2. **Flipped** $N_f(\mathcal{A})$: Flipped descriptions include the base descriptions with an additional description of all predicates that are deleted preconditions in that action schema, that is, for an action schema $a \in \mathcal{A}$, $N_f(a)$ is $N_b(a)$ extended with a description of predicates in $pre(a) \cap del(a)$

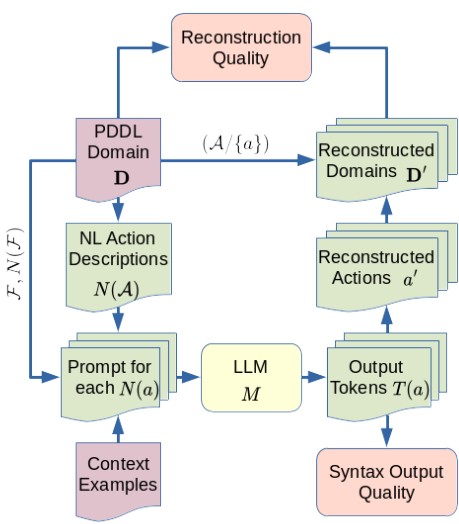

Figure 1: A high level overview of our proposed task

as preconditions. The motivation behind this class is to evaluate if predicates that are explicitly changed are the most important things to include in a natural language description for the LLM, as they might be for a person when describing a domain. For example: *"The action 'unstack' will have a hand unstack a block x from a block y, if the block x is clear, on top of y, and the hand is empty."*

3. **Random** $N_r(\mathcal{A})$: Random descriptions act as a random baseline to compare against flipped descriptions, as well as another higher information content baseline to compare against base descriptions. For each action schema $a$, the description includes the base description $N_b(a)$, and descriptions of $|pre(a) \cap del(a)|$ random predicates sampled from $pre(a), add(a)$ and $del(a)$, where is the description is dependent on if the predicate was sampled from the precondition or effect. For example: *"The action 'unstack' will have a hand unstack a block x from a block y, if the hand is empty and x is on y. After the action, y should be clear."*.

### Evaluating

When considering how to evaluate the performance of LLMs on this task, note that LLMs will frequently output sequences of tokens for our evaluation that cannot be interpreted as a valid planning domain. Some of these errors are syntax based while others are based on the semantics of the underlying PDDL tokens. If a model does output a valid domain, it must be evaluated in terms of its quality.

### Domain Reconstruction Quality Metrics

Evaluating the quality, a correctly generated planning domain is a difficult task. Current metrics such as human expert evaluation (Guan et al. 2023; LI et al. 2023; Hayton et al. 2020) provide a rough but subjective measure that is impossible to automate. Like Guan et al. (2023), we have designed our task such that all generated domains are based off

an existing domain which we can evaluate with respect to a baseline. We look at and evaluate two automated metrics for measuring the quality of generated domains. The first metric, action reconstruction error, is a more traditional automated metric that measures the distance between generated actions in domains, but we note it is a poor metric. We propose a second metric, heuristic domain equivalence, which provides a more robust and tolerant approximation of true domain equivalence.

**Action Reconstruction Error (ARE)**    The *Action Reconstruction Error* (ARE) is a measure of how different two action schema $a, a' \in \mathcal{A}$ are. We define the action reconstruction error as the size of the difference of predicates in the precondition and effect between $a$ and $a'$:

$$\text{ARE}(a, a') = |pre(a) \triangle pre(a')| +$$
$$|add(a) \triangle add(a')| +$$
$$|del(a) \triangle del(a')|$$

where $A \triangle B$ is the symmetric difference $(A/B) \cup (B/A)$. This metric is useful for understanding the distribution of how close the domains output by the models with respect to the original domains. However, we claim that this metric is not a good measurement for actual domain quality. It does not take into account the fact that preconditions and effects can be added or removed from an action without changing the meaning of the action at all, for example, adding a static predicate from a precondition as an effect. To remedy this, we propose an alternative metric based on how usable the domain is for planning.

**Plan Applicability for Heuristic Domain Equivalence** The primary reason a planning domain is created is so that it can be used as the underlying representation for a set of problems in the domain. The problems implicitly define a set of plans, and when reconstructing domains, we can measure domain equivalence in terms of equivalence of the sets of plans for a collection of problems. While it is not practical to check if the full set of plans is equivalent, it is possible to check for a number of plans on number some problems we care about in the domain.

The domain equivalence heuristic is computed as follows: given an original planning domain $\mathbf{D}$, a reconstructed planning domain $\mathbf{D}'$, and a set of solvable planning problems for $D$, $\mathbf{P}_D$, each problem $\Pi \in \mathbf{P}_D$ can be transformed into a problem $\Pi' \in \mathbf{P}_{D'}$ that uses $\mathbf{D}'$ as its underlying domain. For each such pair of problems $\Pi$ and $\Pi'$ and some corresponding subsets of their plans $P \subseteq \mathcal{P}_\Pi$ and $P' \subseteq \mathcal{P}_{\Pi'}$, we can cross check whether $P \subseteq \mathcal{P}_{\Pi'}$ and $P' \subseteq \mathcal{P}_\Pi$. For each individual plan, the test can be efficiently performed using a plan validator[1]. This heuristic, plan equivalence on $\mathbf{P}$ for a subset of plans, is a necessary condition for true domain equivalence, and its negation is a sufficient condition to show true domain inequality.

**Result Classes**
We propose four result classes for classifying the action from an LLMs output. Each class other than the heuristically

---

[1]https://github.com/KCL-Planning/VAL

equivalent domain class has multiple sub-classes to give a better idea of the types of problems encountered.

1. **Syntax Error**: The model produced syntactically invalid PDDL. This PDDL cannot be parsed to evaluate an action reconstruction error with. Subclasses (in precedence order): (1) No PDDL (NoPDDL): Model did not output any PDDL, (2) Parenthesis Mismatch (PError): issues regarding the matching parenthesis in the PDDL (3) Unexpected Token (UToken): The PDDL parser failed after finding an unexpected token.

2. **Semantic Error**: The model produced syntactically valid PDDL, but the PDDL doesn't integrate with the intended problems. Subclasses (1) Type Error (TError): The model produced an unexpected type (2) Predicate Argument Error (PAError): the wrong number of variables were passed to a predicate (3) Wrong Action Name (NError), The name of the action is wrong (4) Bad Precondition (BPError): PDDL STRIPS does not allow negated preconditions, but one is present.

3. **Different Domain**: The model produced syntactically valid PDDL that integrates with the original domain, but the underlying domains are different by way of the domain equivalence heuristic. The behavior of the actions is not as intended, plans from the original domain cannot be applied in the new domain and vice versa. Subclasses (1) No Plans Found (NoPlan): No plans were able to be found on problems in the new domain (2) New Plan Application Error (NPApp): Could not apply a new plan to the original domain (3) Original Plan Application Error (OPApp): The original plan could not be applied to the new domain.

4. **(Heuristically) Equivalent Domain**: The model produced syntactically valid PDDL that integrates with the desired domain under the domain equivalence heuristic, plans from the original domain can be applied in the new domain and vice versa.

The classes form a hierarchy in which syntax errors superseded semantic errors which supersede both the different and equivalent domain classes which are mutually exclusive. *i.e.* An output with both syntax and semantic errors will only be marked as a the error caught first, the syntax error.

## 4   Experiments and Results
### Setup

For evaluation, we evaluate over the LLaMA family of LLMs (Touvron, Lavril, and Izacard 2023), as well as Star-Coder (SC) (Li, Allal, and Zi 2023). For LLaMA we evaluate over both the base pre-trained models at 7b, 13b, 70b parameters. We also evaluate the 7b, 13b, 70b LLaMA models that have been fined tuned for chat using reinforcement learning with human feedback (RLHF) (Ouyang, Wu, and Jiang 2022). For token selection for all models, we use greedy sampling in which the token with the highest output probability is selected as the next token.

For our domains we select 9 PDDL domains with varying action and predicate complexities. We include 4 domains

from (Silver et al. 2023) guaranteed not to be in the training set, as they were created after LLaMA and StarCoder were trained. These domains are marked with a†. The remainder of our domains are famous classical planning domains, many of which have appeared in International Planning Competitions.

1. Blocksworld – 5 predicates 4 actions : A robot hand tries to stack blocks on a table in a particular configuration.

2. Gripper – 4 predicates 3 actions : A robot moves balls from one room to another using grippers.

3. Heavypack†– 5 predicates 2 actions : Specified items must be packed into a box depending on item weight.

4. Hiking†– 5 predicates 2 actions : Hikers must navigate to a location over varying terrain.

5. Logistics – 3 predicates 6 actions : Items must be transported to locations using planes and trucks.

6. Depot – 6 predicates 5 actions : A combination of blocks and logistics domains.

7. Miconic – 6 predicates 4 actions : A lift delivers multiple passengers to their desired floors from their starting floors.

8. Trackbuilding†– 4 predicates 3 actions : An agent must build a path for a train to take to a given location.

9. TrapNewspapers†- 7 predicates 3 actions : A deliveryperson must deliver newspapers to a number of safe locations from a home base.

For the domain equivalence heuristic, our problem set consists of 2 simple randomly selected problems from each domain. We select 10 plans using a top-k planner $K^*$ (Lee, Katz, and Sohrabi 2023). The top-k plans for a problem $\Pi$ are the set of $k$ different plans with the lowest costs, where in our case of unit costs is the same as the length of the plan. While any $k$ plans could be used for computing the domain equivalence heuristic, using the top-k plans we ensure that minimally the optimal plans for the evaluated problems are equivalent. To test for plan validity we use VAL.

**Evaluating Heuristic Domain Equivalence Over Different LLMs**

For this experiment we exclusively use base descriptions in which only a description of the action's parameters and types without reference to predicates. For prompt generation, each base action description is turned into 60 prompts, each with 3 randomly sampled context examples from outside of its domain. We note that this sampling is done uniformly across all types of actions, the only restriction being that the action used for context cannot be in the same domain as the action we are generating for. We chose to use 60 prompts as a trade-off between experiment runtime and statistical significance. We chose to use 3 context examples after a manual parameter search, finding increasing the number of context examples further did not improve results, while decreasing past 3 lead to worse results.

Figure 2 (Top) displays the breakdown of outputs over the primary result classes. Two results are immediately apparent from this. First, LLMs particularly larger ones, are

| Result | SC | 7b | 7bC | 13b | 13bC | 70b | 70bC |
|---|---|---|---|---|---|---|---|
| Syntax | 3.70 | 15.31 | 22.03 | 1.30 | 25.73 | **0.36** | 8.49 |
| NoPDDL | 0.00 | 0.00 | 0.00 | 0.00 | 0.00 | 0.00 | 0.00 |
| PError | 0.31 | 0.21 | 0.16 | **0.00** | 0.10 | **0.00** | 0.05 |
| UToken | 3.39 | 15.10 | 21.82 | 1.30 | 25.62 | **0.36** | 8.07 |
| Semantics | 18.02 | 22.29 | 36.15 | 14.64 | 22.97 | **7.08** | 11.72 |
| PAError | 15.57 | 17.55 | 25.10 | 8.59 | 15.26 | **4.58** | 9.17 |
| NError | 0.16 | 0.10 | **0.00** | 0.05 | 0.16 | 0.10 | 0.05 |
| TError | **2.29** | 4.64 | 10.57 | 5.78 | 7.45 | 2.40 | 2.50 |
| BPError | **0.00** | 0.21 | 0.47 | 0.21 | 0.10 | **0.00** | **0.00** |
| Diff | 67.55 | 56.46 | **36.25** | 75.21 | 43.13 | 63.75 | 58.07 |
| NoPlan | 51.72 | 44.64 | **23.07** | 59.43 | 26.72 | 42.50 | 41.77 |
| NPApp | **7.76** | 8.85 | 8.80 | 8.96 | 11.67 | 12.34 | 11.20 |
| OPApp | 8.07 | **2.97** | 21.72 | 6.82 | 4.74 | 8.91 | 5.10 |
| Equiv | 10.73 | 5.94 | 5.57 | 8.85 | 8.18 | **28.80** | 21.72 |

Table 1: Distribution of Result Classes and Subclasses. Lower is better for all classes and subclasses except equivalent domain (Equiv), for which higher is better. Best results in bold.

quite good at generating syntactically and semantically valid PDDL, the best model LLaMA-2-70b, is able to construct valid PDDL in 93% of domains. When looking at valid PDDL generated, we see that the ratio of heuristically equivalent domains to non-equivalent domains and number of heuristically equivalent domains is largely dependent on model size (see Figure 3). The best result was on LLaMA-2-70b. It reconstructed 29% domains to be heuristically equivalent to the natural language descriptions. This is a very promising result in terms of the applicability of LLMs for the task of PDDL domain generation. Second, in terms of different types of models, it is surprising that the LLaMA chat models perform worse on this task than base LLaMA models across the board. Typically these models that have been trained with RLHF are seen to do better than base models across the board (Ouyang, Wu, and Jiang 2022).

We next turn to discussing result subclasses. Table 1 displays the lopsided breakdown of syntax and semantic errors. There were no instances of the No PDDL subclass, all models evaluated output something minimally interpretative as PDDL. Parenthesis mismatch errors were also negligible. The overwhelming majority of syntax errors were unexpected token errors encountered within the PDDL while parsing which encompassed a whole range of issues from duplicate ":precondition" tags to attempting to add type annotation to variables mentioned in predicates. For semantic errors, the primary breakdown was dominated by issues related to predicate argument counts where the model added or removed arguments to predicates in the action schema. Type errors were rare, we note that StarCoder performed best in this regard. Incorrect action name errors were exceedingly rare. Finally, different-domain subclasses displayed in Figure 2 (Bottom) reveal an interesting insight into the quality of generated domains. The results show that

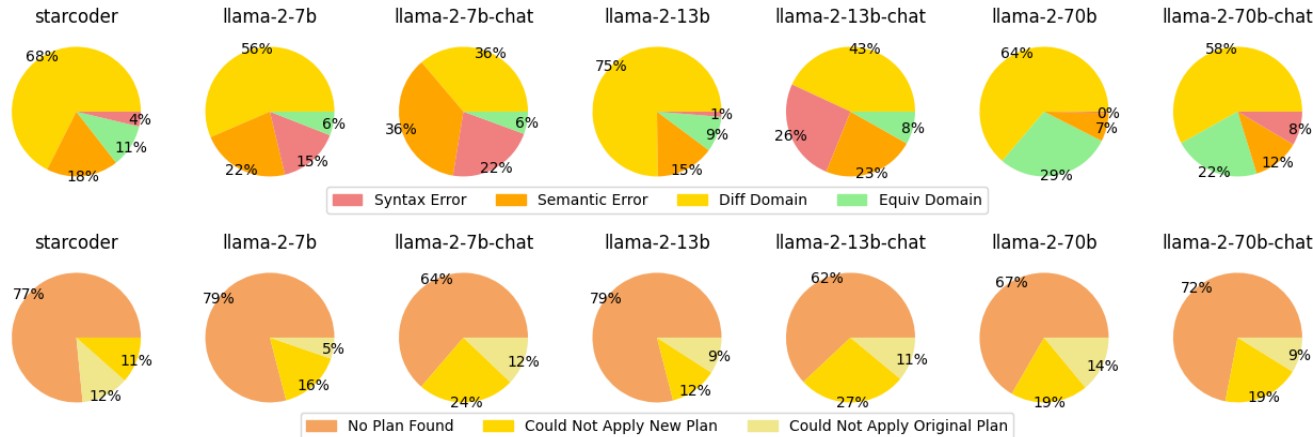

Figure 2: (Top) Characterizing LMM outputs in terms of core result classes. (Bottom) Breakdown of Different-Domain Sub-classes

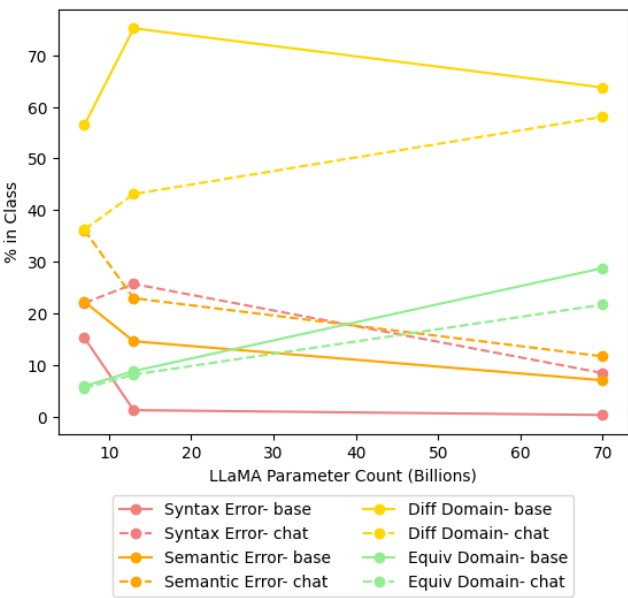

Figure 3: Overview of LLaMA result class percentages with respect to model size. Contains both chat and base models

across the board, the majority of valid generated domains in the different domain result class are not able to be used for planning, with the planner failing to produce any valid plan using the reconstructed problems in domain $P_{D'}$. with the rest falling relatively equally between failing to apply plans. The remaining different-domains failures are split relatively equally due to failures in cross validating the new plans on the original domains and vice versa.

## Evaluating Heuristic Domain Equivalence Over Description Classes and LLMs

For this experiment, we evaluate result classes over the three proposed description classes. To generate our prompts, we map each action to 20 prompts in each of the 3 description classes. The context for the prompts is taken from the same description class and is always taken from domains outside the domain of the action to evaluate. For evaluation we use the same setup as our first experiment and evaluate over our result classes. Figure 4 displays a breakdown of the performance of each model on each description class. The results show that while on some models the flipped class performs well, it is not consistent and not as statistically significant as we had predicted. We are surprised to see that the base class performs on par with the random and flipped classes on the LLaMA models, leading us to conclude that at least for the classes we looked at where the number of predicates in flipped is small, the extra information provided by the random and flipped descriptions is not significant enough to sway the results for these models. The anomaly here is Star-Coder in which providing the extra context in the random and flipped classes boots its performance by around 10%.

## Action Reconstruction Error and Result Class

For this experiment we evaluate the models in terms of their action reconstruction error to see how close from a predicate-by-predicate point of view the model gets to re-constructing the original actions. Additionally, we investigate the the relationship between the action reconstruction error and the result classes as well as how the action reconstruction error may be used to augment our use of heuristic domain equivalence. This experiment uses the same setup as the experiment over description classes, each $a \in \mathcal{A}$ is mapped to $N_b(a)$ and is used for 60 prompts. All prompts are evaluated on each LLM and result classes and ARE is evaluated for classes for all classes except syntax errors as ARE cannot be automatically computed without a parsed action.

Figure 5 displays the distributions of action reconstruction errors (ARE) for each model, and splits each bucket by reconstruction class. This gives a good picture of how much each model deviates from the original action. We note that the better performing models tend to have their distributions

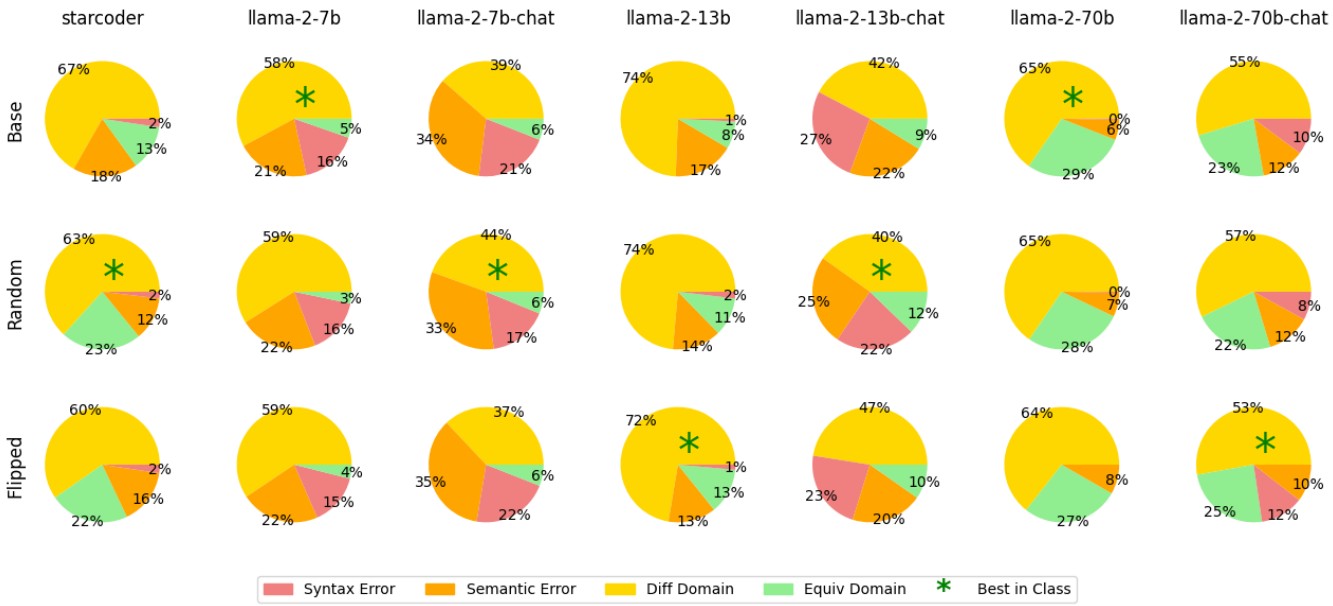

Figure 4: Breakdown of LLMs over Top Level Result Classes vs Different Description Classes.

cluster around lower AREs, that is, they construct actions that are similar in terms of the exact predicates used in the original action. This additionally exposes the flaws of ARE as a metric for domain equivalence as we can see that just being close to the original action in terms of predicate similarity is not good enough and that plenty of domains outside this range are heuristically equivalent. This understanding of ARE can also help us find false positives in heuristically equivalent domains that are not truly equal, since only a finite number of problems and plans for each problem can be evaluated. Hence when searching for false positives it can be useful to start with domains with the highest ARE since it is more likely something with many predicates changed from the original action represents a different domain.

## 5   Related Work

**Large Language Models and Planning**

There are been a number of papers that investigate the use of LLMs for planning. Some recent work (cf. Valmeekam et al. (2022); Raman et al. (2022)) use LLMs as planners, while others (cf Guan et al. (2023); Liu et al. (2023)) use LLMs as auxiliary components of a hybrid planning system while leveraging automated planners for solving the planning task. The general consensus seems to be that LLMs are not very good as planners. This finding was one of the motivations for this work in this work, as we focus on using LLMs to aid automated planning rather than as planners themselves.

**LLM+P**   The LLM+P framework (Liu et al. 2023) was one of the first to recognize the potential of combining LLMs and planners as hybrid systems, and utilizing LLMs to east the use of automated planners . The LLM+P architecture takes in (1) natural language descriptions of problem in a planning domain, (2) a context example of a natural lan-

guage problem in the given domain being converted to a PDDL problem, and (3) a PDDL domain file. Using these inputs the model uses an underlying LLM to convert the natural language problem description and context into a PDDL problem. This is then combine with the PDDL domain input to an automated planner producing a PDDL plan, the resulting plan is then fed into an LLM which describes the plan in natural language. LLM+P's applicability is somewhat hindered by their assumptions that a PDDL domain exists, and context examples converting natural language descriptions of problems to PDDL problems for these domains exist. Such assumptions are impossible to meet in the case of things like narrative action model acquisition, and indeed still requires an expert in the system somewhere to write the domains and the context examples. Our work does not focus on using LLMs to generate PDDL tasks, but it is tangential to all of LLM+P's assumptions. We (1) investigate the construction the PDDL domain rather than have it provided and (2) do this using context examples from arbitrary domains rather than from the same domain.

**LLM-DM**   The most closely related work to ours is the end-to-end domain construction and planning framework from Guan et al. (2023) which we will call LLM-DM. LLM-DM is composed of a three-part process, automated domain construction, human refinement of domain, and planning with the domain. We are interested primarily in their automated domain construction as it is a very similar task to ours. For this, LLM-DM generates a domain on an action-by-action basis, each prompt containing five parts: (1) an instruction describing the PDDL creation task, (2) one or two context examples from the blocksworld domain on what a translation of an action description to PDDL looks like, (3) a natural language description of the domain, (4) a natural language description of the action and (5) a dynamically up-

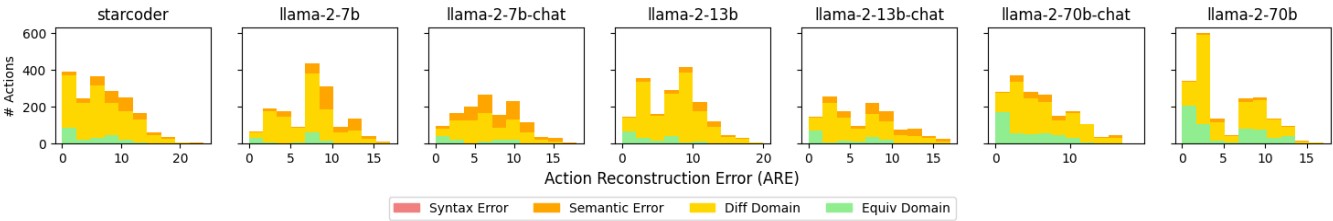

Figure 5: Action Reconstruction Error (ARE) Distribution with respect to Reconstruction Class Over LLMs

dated list of predicates used by the domain including natural language action descriptions. As the domain is generated action-by-action, the instruction and context examples include requests for the model to generate a list of new predicates based on the description of the action. LLM-DM evaluates constructing PDDL on three domains (Logistics, Tyreworld, and a custom domain, "Household") using the LLMs GPT-4 (OpenAI 2023) and GPT-3.5 Turbo (ChatGPT). To measure the quality of the constructed domain, manual human evaluation is used, experts annotate the PDDL domain output, marking the PDDL with mistakes and corrections, which the authors claim provides and approximate distance between the generated PDDL and correct PDDL.

LLM-DM provided inspiration in our work to generate domains using LLMs on an action-by-action basis rather than trying to have the LLM output the full domain. The authors cite well-founded concerns about the context window size and the potential for corrective feedback on an action-by-action basis, making this more useful for the end user. For our work, instead of providing the model with a description of the domain and having the model extract the predicates at each stage on-top of the action translation, we explicitly provided the allowed predicates and their description *as* the description of the domain. This change is key for being able to automatically evaluate the constructed domains, and is responsible for our automated evaluation approaches rather than a manual evaluation approaches.

## Textual and Narrative Action Model Acquisition

The task we propose is similar to the action-model extraction from text task (Lindsay et al. 2017) and narrative action-model acquisition task from text task (Hayton et al. 2020; LI et al. 2023) in which the goal is from natural language to generate the entire domain model from $\mathcal{F}_g$ and $\mathcal{A}_g$ if grounded and $\mathcal{F}, \mathcal{A}$, and potentially $\mathcal{C}$ if lifted. A downside of these tasks is that it very difficult to automatically evaluate performance on, as it requires a full understanding of the natural language text and expert knowledge of PDDL domains. Evaluation for these tasks is frequently done either via expert analysis of the generated PDDL domain such as in (Hayton et al. 2020; Huang, Chen, and Zhang 2014) or automated metrics such as that can't fully capture the performance of the model. These shortcomings in evaluation were a driver of both our problem formulation and proposed domain quality metrics.

## 6 Conclusion and Future Work

There are many avenues which could be explored using this work as a springboard. In particular we are interested in three main directions: (1) deeper investigations of the capabilities of large language models in terms of selection and tuning, (2) using re-prompting for fixing mistakes in PDDL for chat-based LLMs, (3) investigating more robust tasks and metrics.

First, in terms of LLMs there is a lot that could be done to extend this work. The results showing improved performance on larger models is a good starting point for future work and is in line with Guan et al. (2023) which evaluates with respect to GPT-4 and GPT 3.5. coming to similar conclusions that larger pre-trained models are better when it comes to handling PDDL construction. Future work and applications not interested in tuning should take this into consideration using larger models such as GPT-4 and LLaMA-70b as baselines, other large models such as Bloom (Big-Science Workshop 2022) would be promising to evaluate over. Our experiment over description classes revealed the coding model StarCoder performs quite well in certain cases when additional predicate information is included in natural language descriptions, we believe this warrants a further investigation of coding models and their capabilities. Beyond just selection of LLMs, there are two more properties of LLMs we could investigate. First, LLM tuning approaches, such as fine tuning and prompt tuning have been shown to allow small LLMs to perform well on tasks they are tuned on. Second, chat based LLMs with large context windows have can be re-prompt and provide corrective feedback (Raman et al. 2022). Guan et al. (2023) use successfully demonstrate corrective reprompting from tools like VAL and other reprompting to provide corrective feedback to LLMs. Using our result classification system, adding support for corrective reprompting where the re-prompt is based on information regarding result class is a clear next step.

Finally, we discuss a potential alternatives that could be made to our evaluation. As discussed in our approach, we do not use $\mathcal{A}'$ as the set of action schema for a $\mathbf{D}'$ for a number of practical reasons. However, evaluating the performance of domains in which all actions are generated is desirable target for evaluation. Towards this end, it would be interesting to evaluate with respect to a form of iterative domain completion task after an initial action has been generated. Previously generated actions in $\mathcal{A}'$ could then be used as part of the prompt until a full reconstructed action schema for the reconstructed domain $\mathbf{D}'$ has been constructed.

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
