# OpenReview forum: "Large Language Models as Planning Domain Generators"
_icaps-conference.org/ICAPS/2024/Conference — ICAPS 2024_

### Official Review · Reviewer_yoNB · 2024-01-17

**Significance And Importance:** 2
**Soundness:** 3
**Novelty:** 2
**Clarity:** 3
**Overall Evaluation:** 1
**Confidence:** 3

**Weaknesses:**

0: Minor weaknesses requiring some work to be addressed for the paper to be accepted.

**Contributions Of The Paper:**

The paper addresses the challenge of manually creating planning models, specifically domain models in the field of AI planning. The objective is to simplify this process by exploring the use of large language models (LLMs) to automatically generate planning domain models from textual descriptions. The authors introduce a task and an automated evaluation method that involves comparing sets of plans for domain instances. The study includes an empirical analysis of seven large language models, encompassing both coding and chat models, across diverse planning domains. The findings indicate that LLMs, especially larger ones, demonstrate a certain level of proficiency in generating accurate planning domains from natural language descriptions.

**Ethical Considerations:**

(1) Not Applicable: The paper does not have any ethical considerations to address

**Nomination For Best Paper:**

No

**Questions For Authors:**

Do we have the option to utilize a more abstract level for LLMs to generate PDDL domains, reducing the need for extensive task-specific information?

Can the approach integrate newly acquired information into the PDDL? For instance, in the scenario where a robot explores its surroundings and identifies previously unknown objects, will these objects be added to the domain automatically?

**Reproducibility:**

3: Authors describe the implementation and domains in sufficient detail.

**Strengths Of The Paper:**

The paper delves into an intriguing AI planning domain generator, employing experiments to evaluate the efficacy of the proposed approach.

**Weaknesses Of The Paper:**

It appears that we are required to provide an excessive amount of information to LLMs to generate planning problems.

---

> ### Author Rebuttal · Authors · 2024-01-27
>
> **Q1**
> We use task specific information (predicates and their natural language descriptions from the base domain, action names, etc)
> for automatic domain evaluation **only**. Without the task specific information, it would be impossible to automatically evaluate the quality of reconstructed domains due to the possibility that the LLM would select novel action or predicate names.
> However, in practice, this level of detail can be relaxed when generating a novel PDDL model from LLMs (as use of same predicates, or action name would not be a requirement for novel domain).
> Further, at least some of this information can in turn be obtained, e.g., with LLMs and doing so is a promising future direction.
>
> **Q2**
> In this work we focus on learning lifted action models. Identifying unknown objects will change the problem file, which we assume to be provided in this work and is handled by existing work [1].
> More related to our setting is a possible realization that a provided set of predicates is insufficient, and a predicate should be added to the PDDL model, relaxing our assumption on the provided input. This is one of the more promising avenues for future work.
>
> [1] Liu et al, 2023, [LLM+P: Empowering Large
> Language Models with Optimal Planning Proficiency](https://doi.org/10.48550/arXiv.2304.11477)

---

### Official Review · Reviewer_7er5 · 2024-01-22

**Significance And Importance:** 2
**Soundness:** 3
**Novelty:** 2
**Clarity:** 3
**Overall Evaluation:** 1
**Confidence:** 4

**Weaknesses:**

0: Minor weaknesses requiring some work to be addressed for the paper to be accepted.

**Contributions Of The Paper:**

The paper makes four key contributions:

-- Task Definition for PDDL Domain Reconstruction: They define a novel task of reconstructing PDDL domains from natural language descriptions. This task is based on ground truth, which involves creating PDDL domains that match a pre-existing, validated domain.
-- Automated Evaluation Metrics: They introduce two new metrics for evaluating the quality of the generated PDDL domains. Notably, these metrics do not rely on manual human evaluation, marking a significant step towards automating the evaluation process.
-- Analysis of Natural Language Descriptions for PDDL Actions: The study investigates various classes of natural language descriptions of PDDL actions. This examination aims to understand how including or excluding specific information affects the ability to generate PDDL domains and the quality of the domains generated.
-- Empirical Evaluation of Different LLMs: They conduct an empirical analysis of seven large language models (LLMs), including coding and chat models. The study provides a detailed analysis of the performance of these models across nine different planning domains​​.

**Ethical Considerations:**

(1) Not Applicable: The paper does not have any ethical considerations to address

**Nomination For Best Paper:**

No

**Questions For Authors:**

1) Clarify how the methodology accounts for potential inaccuracies or ambiguities in the natural language descriptions of PDDL actions. What strategies are employed to ensure that these descriptions accurately reflect the intended actions?
2) How does the methodology handle complex domains where the relationships and interactions between actions are highly intricate? Are there any limitations in the current approach when dealing with such complexities?
3) How well do the models generalize to entirely new or significantly different domains?
4) Please elaborate on the limitations of the newly introduced automated evaluation metrics. Are there aspects of domain quality that these metrics might not capture, particularly in terms of functional correctness in practical scenarios?
5) Please address the challenges mentioned in the weakness part, posed by the limited context window of LLMs, especially in the case of large and complex domains. How does this limitation affect the model's understanding and generation of domain models?
6) How does the reliance on pre-defined predicate descriptions impact the applicability of the methodology in situations where such descriptions are not readily available?
7) Please comment on the computational intensity of using large language models. What are the implications for accessibility, especially for users with limited computational resources?
How do these domains perform in practical planning scenarios beyond theoretical evaluations?

**Reproducibility:**

1: Difficult to reproduce because of missing detail.

**Strengths Of The Paper:**

-- Innovative Task Definition: The paper provides the task of reconstructing PDDL domains from natural language, which can be considered as a  significant advancement as it opens up new avenues for automating the creation of planning models, which traditionally require extensive manual effort.
-- Automated Evaluation Metrics: The development of two new metrics for evaluating the quality of generated PDDL domains without relying on manual human evaluation is a notable strength. These metrics enhance the objectivity and scalability of the evaluation process.
-- Comprehensive Analysis of Natural Language Descriptions: The authors examine various classes of natural language descriptions for PDDL actions, and how the inclusion or exclusion of certain information impacts domain generation. This analysis deepens understanding how NL is interpreted and transformed into structured planning models.
-- Empirical Evaluation Across Multiple Models: The study's evaluation of seven different LLMs including coding and chat models across nine planning domains provides a thorough and comparative analysis.
-- Action-by-Action Domain Reconstruction: The methodology of reconstructing domains on an action-by-action basis is a practical approach that mitigates the challenges posed by the limited context window of LLMs. This methodical approach allows for more focused and manageable domain generation.
-- Ground Truth-Based Approach: The use of ground truth in the task definition enhances the reliability and relevance of the generated domains, ensuring that they are not only syntactically correct but also practically usable in real-world planning scenarios.
-- Broad Applicability: The research has broad implications for the field of AI planning, as it suggests a way to leverage the power of LLMs for automating one of the most labour-intensive parts of planning, i.e., the creation of domain models.

**Weaknesses Of The Paper:**

General approach:
-- Dependence on Accurate Natural Language Descriptions: The methodology relies heavily on accurate natural language descriptions of PDDL actions. Any ambiguity or inaccuracy in these descriptions could significantly impact the quality of the generated PDDL domains.
-- Potential for Overfitting to Known Domains: Since the methodology involves training on known domains, there is a risk that the models may become overfitted to these domains and may not generalize well to entirely new or significantly different domains.
-- Evaluation Metrics Limitations: While the paper introduces new automated evaluation metrics, these metrics might not capture all aspects of domain quality. For example, they might not fully account for the functional correctness of the generated domains in practical planning scenarios.
-- Reliance on Pre-Defined Predicate Descriptions: The approach assumes that a list of predicates and their descriptions are provided. This reliance could limit the approach's applicability when such descriptions are unavailable or difficult to generate.

Limitations regarding LLM usage:

-- Domain Context Understanding: The approach assumes that the LLMs understand the domain context well, which may not always be the case. LLMs can sometimes generate plausible but incorrect or irrelevant content, especially in more complex or less common domains.
-- Challenges with Complex Domains: The methodology may struggle with complex domains where the relationships and interactions between actions are intricate and nuanced. LLMs might not always capture these subtleties effectively.
Context Window Limitations: The limited context window of LLMs can be challenging, especially for large and complex domains. This limitation might lead to an incomplete or fragmented understanding of the domain by the LLM.
-- Computational Resources: Using large language models can be computationally intensive, which might limit the approach's accessibility for individuals or organizations with limited computational resources.

---

> ### Author Rebuttal · Authors · 2024-01-27
>
> 1. **Ambiguity:** We considered 3 description classes with varying amounts of information about original actions (Sec. 3), including where no preconditions or effects are provided in the description. These classes can be seen as a proxy for ambiguity. Evaluations in Fig. 4 with the description class do not show significant impact in performance. **Inaccuracy:** The burden of ensuring that the description reflects the intended actions is on the end user. For our dataset, we performed manual review to ensure that the descriptions do match.
> 2. Preliminary evaluations indicated negative correlation between the domain complexity & ability of LLM to generate heuristically equivalent PDDL.
> 3. We included 4 entirely new domains in our experiments: Heavypack, Hiking, Trackbuilding & TrapNewspapers. If we break the results down by domain, some of the best results are in these domains.
> 4. ARE is prone to false negatives, as actions do not have to share preconditions & effects to be equivalent. HDE has false positives but is pragmatic (See response to 7MVb). It enables evaluation wrt actual problems for the domain, and is thus suited for practical planning scenarios. By evaluating on larger number of diverse problems, HDE provides certainty about domain equivalence that ARE does not.
> 5. We circumvent the context window limit by translating one action at a time. Our largest prompt was ~1000 tokens, while the context window was 4096 tokens. In practice this would not be an issue, as errors from the domain complexity would compound and lead to poorly generated domains way before the context window is reached.
> 6. The pre-defined predicate descriptions are not strictly required for evaluation with our method. We will make this clear in the paper. Even without it, the LLM could still infer meanings of predicates from predicate names, parameters, and in-context usage examples, and would likely be able to reconstruct some domains, albeit with some impact on performance.
> 7. **Accessibility:** We evaluate LLM without fine-tuning. Most workstations can run the LLaMA 7b models. All of our experiments can also be run using free services (like Huggingface) in a week. **Practical problem:** We test on PDDL domains with varying action/predicate complexities. While application domains may have larger number of actions, each action is usually of similar complexity. Since we construct PDDL action-by-action, the amount of actions should not matter much.

---

### Official Review · Reviewer_7MVb · 2024-01-23

**Significance And Importance:** 3
**Soundness:** 4
**Novelty:** 2
**Clarity:** 4
**Overall Evaluation:** 2
**Confidence:** 3

**Weaknesses:**

1: Minor weaknesses that are easily fixable.

**Contributions Of The Paper:**

This paper aims to evaluate LLM’s abilities to generate PDDL domains from textual descriptions (natural language descriptions). The task of evaluating the usefulness of the generated domain description is difficult. Previous works leveraged human experts for evaluation.
Authors present a novel PDDL task for domain reconstruction from natural language (based off a ground truth) and two metrics for evaluating domain quality (Action reconstruction error and Heuristic domain equivalence). Additionally, they investigate if adding or removing information of classes of natural language descriptions of PDDL actions impacts the ability to generate or the quality of the generated domains and an empirical analysis of 7 LLMs across 9 different planning domains.
To convert PDDL action schema to their natural language description, they present three different strategies. Base N_b(A), consisting of the action name, parameters, and the parameter types of action, a one-line description of what the action does without mentioning explicitly any predicates. Flipped N_f(A), including the base description with an additional description of all predicates that are deleted preconditions in that action schema. Random N_r(A), including the base description and descriptions random predicates sampled from pre(a), add (a) and del(a).
For evaluating the domain reconstruction quality, authors suggest ARE (action reconstruction error) that measures the distance between the generated actions in domains and the heuristic domain equivalence that measures in terms of equivalence of the sets of plans for a collection of problems how usable the domain is for planning.
Authors use a hierarchy of four classes for classifying the action from a LLM output.
1.	Syntax error
2.	Semantic error
3.	Different domain
4.	Heuristically equivalent domain
From the results, they show that larger LLMs exhibit some effectiveness in generating correct planning domains from natural language descriptions. However, most valid generated domains are not able to be used for planning. Also, there was no significant enough difference among the three different strategies used to evaluate the impact, when adding or removing information, in the ability to generate or the quality of the generated domains.

**Ethical Considerations:**

(1) Not Applicable: The paper does not have any ethical considerations to address

**Nomination For Best Paper:**

No

**Questions For Authors:**

Q1. You mentioned that the understanding of ARE could be useful to find false positives in heuristically equivalent domains. Did you carry on any evaluation, or do you have any insights concerning the impact of false positives in the proposed automatically evaluation of generated domains?

**Reproducibility:**

5: Code and domains (whichever apply) are already publicly available

**Strengths Of The Paper:**

The paper is well written, it presents an interesting evaluation that gives insights to the planning community regarding the work with LLMs. The shared resources for this work are likely to be moderately useful to other researchers. Key resources are available and key details (e.g., experimental setup) are sufficiently well-described for competent researchers to confidently reproduce the main results.

**Weaknesses Of The Paper:**

In their work, authors mention that explicitly provide the allowed predicates is key for being able to automatically evaluate the constructed domains. I would have liked a comparison in terms of time or simplification of the evaluation task with state-of-the-art work to get a clearer idea of the advantage this leads to.
Other comments:
Page 5, second column, the end of the sentence “When looking at valid PDDL generated,…” needs to be reviewed.
Page 7, 1st column, the sentence “This finding was one of the motivations for this work…” needs to be reviewed.

---

> ### Author Rebuttal · Authors · 2024-01-27
>
> For any domain that is heuristically equivalent to the original domain, i.e. if a subset of plans from original domain are applicable in the new domain and vice-versa, there are two cases:
>
> * `ARE == 0`,  the original action is indeed equivalent to the reconstructed action, not just heuristically equivalent, thus it cannot be a false positive.
> * ``ARE > 0``:
>     1. The domains are in fact equivalent (For example, if the original domain used predicate 'on' but the generated domain uses 'above' or the generated action has an additional static predicate in the precondition). So the generated domain is a true positive.
>     2. Domains are not actually equivalent but the generated subsets of plans happen to be compatible. In such a case, the generated domain is a false positive. Consider the following example:
>
> ```
> Generated Action:
>  (:action pick
>        :parameters (?b - ball ?r - room ?g - gripper)
>        :precondition (and (at-robby ?r) (at ?b ?r) (free ?g))
>        :effect (and (not (free ?g)) (carry ?b ?g)))
>
> Original Action:
>  (:action pick
>        :parameters (?obj - ball ?room - room ?gripper - gripper)
>        :precondition  (and  (at ?obj ?room) (at-robby ?room)
>              (free ?gripper))
>        :effect (and (carry ?obj ?gripper)
> 		    (not (at ?obj ?room))
> 		    (not (free ?gripper))))
> ```
> Under our current assumption, which restricts the generated domain to the same set of predicates as the original domain, we hypothesize, but did not emperically validate, that the larger the ARE the more likely the case that a heuristic equivalence is a false positive.
> This follows from the following probabilistic argument: with no information about what permutations have been made to the action, we need to assume each permutation has some fixed chance of making the action not equivalent. It is thus clear the more permutations made to the action, the probability the action breaks compounds.

---

### Meta-Review · Area_Chair_Fmm5 · 2024-02-06

**Recommendation:** Accept (Poster)
**Confidence:** 2

**Metareview:**

This authors address the evaluation of LLMs as planning domain model generators.
They define the task of PDDL domain reconstruction from natural language, define metrics for automatically evaluating domain quality, and evaluate current LLMs on this task.

The reviews were generally favorable, and although there was little discussion after the rebuttal period, as metareviewer, I believe the rebuttals tried to address the reviewer questions.

One disappointing aspect of the paper is that it focuses entirely on classical planning, and while it is interesting to what extent LLMs can reconstruct classical domains, it is not clear whether in practice, the ability to reconstruct classical domains fulfills a true need among practitioners making real applications (as opposed researchers focused on writing ICAPS papers).

Another related issue is the lack of discussion of relationships to any other state-of-the-art work on code generation with LLMs. Given the considerable interest in and progress made in LLM-assisted code generation for standard programming languages (e.g., Copilot), a natural question is how PDDL generation is similar to or different from the LLM-assisted code generation tasks which are already widely studied and used.

**Ethical Considerations:**

(1) Not Applicable: The paper does not have any ethical considerations to address